# Research Progress of Alumina-Forming Austenitic Stainless Steels: A Review

**DOI:** 10.3390/ma15103515

**Published:** 2022-05-13

**Authors:** Ling Liu, Cuilin Fan, Hongying Sun, Fuxiao Chen, Junqing Guo, Tao Huang

**Affiliations:** 1School of Materials Science and Engineering, Henan University of Science and Technology, Luoyang 471000, China; 20160264@ayit.edu.cn (L.L.); fclycn@163.com (C.F.); guojq@haust.edu.cn (J.G.); huangtao@haust.edu.cn (T.H.); 2School of Mechanical Engineering, Anyang Institute of Technology, Anyang 455000, China; 3Provincial and Ministerial Co-Construction of Collaborative Innovation Center for Non-Ferrous Metal New Materials and Advanced Processing Technology, Luoyang 471000, China

**Keywords:** AFA steel, precipitate, corrosion resistance, mechanical properties, creep

## Abstract

The development of Alumina-Forming Austenitic (AFA) stainless steel is reviewed in this paper. As a new type of heat-resistant steel, AFA steel forms an alumina protective scale instead of chromia in a corrosive environment. This work summarizes the types of developed AFA steels and introduces the methods of composition design. Various precipitates appear in the microstructure that directly determine the performance at high temperatures. It was found that alloy elements and the heat treatment process have an important influence on precipitates. In addition, the corrosion resistance of AFA steel in different corrosive environments is systematically analyzed, and the beneficial or harmful effects of different elements on the formation of alumina protective scale are discussed. In this paper, the short-term mechanical properties, creep properties and influencing factors of AFA steel are also analyzed. This work aims to summarize the research status on this subject, analyze the current research results, and explore future research directions.

## 1. Introduction

At present, thermal power remains the world’s most significant energy supply, but global warming caused by carbon dioxide cannot be ignored. To improve thermal efficiency and reduce the emission of carbon dioxide, a new generation of advanced ultra-supercritical power plants has been designed to operate at 700 °C/35 MPa [1,2]. Critical components operating at these temperatures must have a good combination of properties, including good corrosion resistance and creep properties and must also be economically viable [3].

For commercially available heat-resistant steels, ferrite Fe-Cr-Al-based alloys have strong oxidation resistance at high temperatures but poor creep resistance due to their open body-centered cubic (BCC) structure. Oxide-dispersion-strengthened (ODS) ferritic Fe-Cr-Al-based alloys and nickel-based alloys meet the requirements of ultra-critical units, but their applications are limited due to excessive nickel content, which is costly. Although traditional austenitic stainless steels with a face-centered cubic structure have improved creep strength at high temperature by alloying, their oxidation resistance are reduced when the temperature exceeds 600 °C, especially in a severe working environment containing water vapour or sulphur.

In 2007, the concept of a new aluminum-containing austenite steel was first reported by the Oak Ridge National Laboratory. It was developed on the basis of High-Temperature Ultrafine Precipitate Strengthening Steel (HTUPS), which combines the advantages of high-temperature ultrafine precipitation reinforcing and forming the Al_2_O_3_ protective scale at a high temperature. Therefore, it has good creep properties and oxidation resistance, especially in an environment containing water vapour [4]. Compared with chromia, the growth rate of alumina is 1~2 orders of magnitude lower than that of chromia at temperatures above 600 °C, and it has higher thermodynamic stability [5,6]. After the performance test found that AFA steels have excellent oxidation resistance and creep strength at elevated temperatures, they were then considered to have a good application prospect in some important environments, such as chemical petroleum, gas power generation, nuclear power, turbine heat exchanger, especially the fourth generation of ultra-supercritical power generation units [7,8,9]. However, aluminum is a ferritic stable element, and the underlying challenge is that the addition of aluminum to iron results in a loss of alloy creep strength by stabilizing the weak ferritic form of Fe at the expense of the stronger austenitic Fe form. Therefore, the dilemma of AFA alloy design is how to balance the forming of alumina protective scale and keep a single austenite structure under relatively economical conditions.

By analyzing the literature, one finds many studies mainly focused on how to adjust the content of alloying elements to achieve excellent oxidation resistance and creep properties. However, there are only a few reports on the applications of AFA steel. It was reported that OC4 (Fe-14Cr-25Ni-3.5Al-2.5Nb) foil was used to make air cells in a microturbine and recuperator, and the performance began to decline after 3000 h at 720 °C. In addition, cast AFA steel was used to make a trial furnace roller, but its test performance has not been reported [10,11,12]. Although AFA steel has attracted significant attention from researchers, there is still a lack of reviews focusing on AFA steel systems [8,9,13,14]. This paper summarized the research status of AFA steel, analyzed its oxidation resistance and creep properties and mechanism, and suggests future research directions for AFA steel.

## 2. The Types of AFA Steel and Design Method

### 2.1. The Types of AFA Steel

Researchers have conducted numerous studies on the above key issues and designed some AFA steels with excellent performances by alloying due to the need to balance mechanical properties and corrosion resistance, the relatively low level of aluminum and chromium used in AFA alloys. Thus, all AFA steels exhibit a transition from the protective alumina scale to the internal oxidation of aluminum (and nonprotective corrosion behavior) if the temperature is raised too high, i.e., they have an upper-temperature performance limit for good corrosion resistance. Based on the content of nickel and the corrosion limit, AFA steels are classified into five categories—typical AFA steels developed are summarized in Table 1.

The first AFA alloy is HTUPS 4 with Fe-14Cr-20Ni-2.4Al wt.%, which forms a continuous and external Al_2_O_3_ protective scale at 800 °C in the air or containing 10% water vapour, and the creep life reaches 2300 h at 750 °C and 100 MPa [4]. On this basis, many AFA alloys were developed by adding different contents of alloying elements according to the target. According to the main function of the alloying elements, it can be divided into basic elements, antioxidant elements and precipitates elements.

The basic elements are iron, nickel, chromium, manganese, and silicon. Iron is the basic element of the iron base alloy, nickel is the basic element to ensure the austenite structure, and chromium is a common element in stainless steel and can promote the formation of alumina and manganese, and silicon can improve the fluidity of the alloy.

The main corrosion-resistant element is aluminum. In addition to aluminum, elements of stabilizing alumina scale are chromium, niobium, carbon and boron, and active elements hafnium and yttrium that improve the adhesion of the oxide scale.

Elements with precipitation mainly refer to the elements that can help the precipitate phase when the alloy is in service. In AFA alloy, niobium, titanium, vanadium, thallium, tungsten, molybdenum, copper, carbon and boron can form a MC or Laves phase or promote a B2 and L12 phase. However, some elements were not unidirectional. The specific classification of elements is shown in Figure 1. The performance impact of these elements is described in more detail in a later section.

### 2.2. The Design Methods of AFA Steels

Research shows that AFA steels with a single austenitic structure have excellent overall properties at high temperatures [4,38]. Thus, the key design strategy is to define compositional guidelines to achieve a protective alumina-scale instead of the Al_2_O_3_-scale on conventional stainless steels, and then maximize second phases precipitate strengthening for high temperatures. According to the comprehensive literature of the past ten or more years, the design methods can be divided into phase diagrams, thermodynamic software and genetic algorithms.

(1)Phase Diagram

Phase diagrams are the most primitive design method, and they must be tested repeatedly to obtain the ideal alloy. The initial composition design of AFA steel was mainly to change the trace components, after a lot of experiments, combined with the method of Fe-Cr-Ni superimposed isothermal phase diagram [4,15,38,48]. Because this method requires a long development cycle and significant experimental funds, it is rarely used.

(2)Thermodynamic Software

Thermodynamic software is the most widely used tool to design alloys at present. The thermodynamic calculation software commonly used in the design of AFA alloy includes ThermoCalc (Stockholm, Sweden) (based on minimized Gibbs Free Energy), JMatPro^®^ (Sente Software Ltd., Surrey Research Park, Guildford, UK) (based on material type) and MatCalc (University of Graz, Graz, Austria). Among them, ThermoCalc is used in a larger proportion.

Y. Yamamoto et al. [42] and Dong et al. [49] designed AFA steels consisting of low nickel and high manganese, or high chromium based on Fe-22Cr-25Ni alloy using ThermoCalc. On the other hand, ThermoCalc can be used to analyze the nitriding depth [50], the volume fraction of precipitates [51] and predict the corrosion resistance of AFA alloy in the liquid lead environment [52].

For other computing software, there is little introduction in the literature. Jang et al. [53] used MatCalc to analyze the driving force of the equilibrium phase and the precipitates of AFA steels. Zhao et al. [25] designed an AFA alloy with 2.8 wt.% Cu by JMatPro^®^. The CALPHAD analysis method is used to predict the volume fraction of precipitates after creep, which provides a microstructure basis for analyzing its high-temperature performance [54].

Although thermodynamic calculation software is widely used in the design of the total composition, it was found that the influence of trace elements on precipitates could not be well-predicted; it is still necessary to provide actual data for optimization calculation [55].

(3)Genetic Algorithm

This method needs to establish a thermodynamic model based on a large amount of data, which is still in the exploratory stage. Previously, Shin et al. [56] analyzed the ten years’ worth of data of AFA steels, discovered the key characteristics affecting the Larson–Miller parameter (LMP) and established a data analysis method combining analysis and machine learning, which can be used as an intermediate tool for new alloys design. Lately, Taymaz Jozaghi et al. [18] developed a new type of alumina forming austenitic stainless steel with the poorest alloy content combining the genetic algorithm and thermodynamic model of CALPHAD method based on HTUPS 4. Although the alloy designed by this algorithm has excellent performance and convenient design, it still needs a lot of research to meet the need for wide application.

## 3. Precipitates of AFA Steel

### 3.1. Precipitates

AFA steels have complicated microstructures, which generally have several types of precipitates both in the matrix and along the grain boundaries (GBs). It is well known that the alloy properties are closely affected by the microstructure of alloys [49], forming conditions [50], and the microstructure evolution.

For the currently developed AFA steels, the precipitates are generally carbides, intermetallic compounds such as the Laves phase, B2 and γ′ phase, and σ and χ phase. The formation of precipitates is related to alloy composition and service temperature. The characteristics of each precipitate are shown in Figure 2.

The precipitate order of these precipitates is relatively fuzzy. In the study with a 20Ni (Fe-20Ni-14Cr-2.5Al-0.9Nb-2Si-0.1C-2.5Mo-2Mn) AFA alloy, it was found [57] that the precipitate sequence at 700 °C was confirmed as follows: NbC (I, II) → NbC (III) + NbC (I, II) + metastable L12(Ni_3_Al) → NbC (I, II, III) + metastable L12 (Ni_3_Al) + M_23_C_6_ → NbC (I, II, III) +M_23_C_6_+Laves (Fe_2_Nb) +B2 (NiAl). In an AFA-Cu alloy, Hongyuan Wen et al. [58] found that M_23_C_6_ and B2-NiAl formed initially at GBs within 1 h, while MC and L12 precipitated in the matrix in a period of 5 to 10 h.

The literature [59] suggests that the difference between precipitate at the twin boundary and normal grain boundary is distinctive at the initial state. A large number of studies have shown that the precipitate sequence and phase composition are related to the alloy composition and service conditions. This will be detailed in Section 3.2.1.

#### 3.1.1. Carbides

There are usually MC and M_23_C_6_-type carbides in AFA steel during aging at high temperatures.

The study found that in AFA steels (20% < Ni_wt%_ < 25%), regardless of alloy composition, MC can precipitate at lower than 1300 °C and higher than 600 °C. The results show that NbC (II) is mainly precipitate in AFA steels containing niobium and carbon due to its good stability and fine size [4,21,28,37,44,60]. As shown in Figure 3, secondary NbC precipitated when AFA steel (Fe-20Ni-14Cr-2.5Mo-2.3Al-2Mn-2.8Cu-0.5Nb) was stretched at the strain rate of 8 × 10^−5^ s^−1^, pinned dislocation and enhanced the strength of the alloy. Since the melting point of NbC (II) is 3600 °C, and its nucleation force is mainly induced by strain [36], most primary NbC (I) of micron-size appear in the melting process, which shows little strengthening effect due to their large size [36,57,61,62]. The volume fraction and size of NbC (II) are affected by aging temperature, aging time, alloy composition and heat treatment [29]—further details will be provided later in this paper.

For AFA steels reinforced by NbC (II), increasing the stability of NbC (II) under a high temperature and reducing the amount of primary NbC (I) is the key to ensuring the high-temperature mechanical properties. A recent study by W.X. Zhao et al. has found that the coarsening of NbC (II) is related to its shape, and spherical NbC (II) has a strong coarsening resistance [61].

M_23_C_6_ is the most important carbide in AFA steel without titanium or niobium, and it is generally Cr_23_C_6_, of which the precipitate temperature is generally 400–950 °C. For AFA steels containing titanium or niobium, a small amount of M_23_C_6_ will precipitate during long-term aging, and in austenitic steel containing titanium, TiC will be partially transformed into M_23_C_6_ after long-term aging. However, in austenitic steels containing Nb, M_23_C_6_ is partially transformed into NbC [57]. In addition, Hongyuan Wen et al. found that M_23_C_6_ is precipitated during aging, and there is cubic-cubic coherent relation with austenite matrix [63].

#### 3.1.2. B2-NiAl

B2-NiAl phase is a kind of intermetallic compound, which is easy to precipitate during aging. The shape of B2 is round or needle-like in different matrices. It was found that the shape is the consequence of different lattice misfits between B2 and matrices [30], and round shape B2-NiAl precipitates only formed in ferrite in the as-rolled sample, which coarsened according to Oswald ripening mechanism [30,64]. In addition, B2-NiAl forms in AFA steels adjacent to the alumina scale at high temperatures, which is often looked upon as the reservoir of Al [16,20,65].

Some studies [22,43,66,67] have found that B2-NiAl has a ductile–brittle transition temperature (DBTT) that is related to the matrix composition, and generally ranges from 500 °C to 800 °C. If the temperature is higher than the DBTT, it would lose its strength effect. For AFA steels reinforced with B2, it is necessary to research the depletion mechanism of B2 and ductile–brittle transition to achieve a balance between oxidation resistance and mechanical properties. This is also one of the research contents of this research group.

In addition, it was found that the orientation relationship between B2 and matrix is not fixed. A result by Geneva Trotter et al. [68] is that there are six variants of the ${\left(111\right)}_{m}{//(011)}_{P}$, ${\left[101\right]}_{m}//\;{\left[111\right]}_{P}$ Kurdjumov–Sachs relationships. However, another result by Hongyuan Wen et al. [63] showed that the B2 precipitated in the coherent grains has a Nishiyama Wasserman (N–W) orientation relationship with the austenite matrix, which is ${\;[\bar{1}12]}_{\gamma }{//[101]}_{NiAl}$ and ${\left[1\bar{1}1\right]}_{\gamma }{//[10\bar{1}]}_{NiAl}$, as shown in Figure 4.

#### 3.1.3. Fe_2_(Mo, Nb)-Laves

The Laves phase is an intermetallic phase whose molecular structure is AB_2_. In AFA alloys, the Laves phase is usually Fe_2_Mo or Fe_2_Nb (melting point ~1900 K). As its nuclear power mainly depends on atomic diffusion, the higher the temperature, the larger the size [37].

The results show that for AFA alloys, the strengthening effect of Laves phase is closely related to the size and distribution. Generally speaking, AFA steels can obtain better creep resistance when the size of Laves phase is stabilized within 100 nm [39,48,69]. In spite of this, the enhancement effect of the Laves phase is also controversial. Zhou et al. [38] found that Laves phase improved the creep properties of Fe-18Cr-25Ni-3Al-0.15Si (wt.%)-based AFA alloy, but it was uncovered that Laves phase had a limited effect on the high temperature creep properties in the study of Fe-20Cr-30Ni-2Nb (at. %) base AFA alloy [39,48]. In addition, it was found that the Laves phase with high thermal stability is more conducive to reducing the creep rate of AFA steels at 750 °C [70]. However, Hu et al. [71] discovered that Laves phase could promote dislocation slip and improve the plasticity of the alloy. The Laves phase on grain boundaries may not negatively affect the ductility of Fe-20Cr-30Ni-2Nb-5Al (at. %) AFA steel.

Similarly, Hongyuan Wen et al. [72] and Andrew Peterson et al. [73] found the Fe_2_Mo-Laves formed in the triple junctions could render the fracture cracks. Thus, the enhancement effect of Laves phase needs to be studied systematically. In addition, it was found that Laves phase has similar coarsening rate constants in different AFA alloys [61,74]. Trotter et al. [75] reported that a moderate mismatch between the Laves phase and austenite matrix helps to maintain a sufficient tensile ductility of the aged AFA steel (Fe–20Cr–30Ni (at. %)). Furthermore, Jundong Jiang et al. [32] found that the phase boundaries between the Laves phase and B2 were an incoherent interface.

#### 3.1.4. γ′ Phase

The chemical formula of γ′ phase is Ni_3_Al, a geometrically close-packed phase (GCP) with L12 structure. The lattice constant is close to that of the austenite matrix, so it is coherent with the austenite matrix. Generally, the γ′ phase formed in AFA steels when nickel content was more than 30 wt.% [39,41,57,60,76]. It was found that there was high coherency between γ′ and austenite matrix [77], and γ′ phase was unstable and would be transformed into B2-NiAl phase in low nickel AFA steels [57]. The addition of more Cu (2.8 wt.%) also promoted the precipitate of Cu-containing γ′. Therefore, more attention should be paid to γ′ reinforced alloys.

In addition, it was found that the coarsening theory is related to its service condition; for example, it coarsens according to Lifshitz–Slyozof–Wagner theory with aging at 800 °C; however, it follows Oswald’s ripening law during creeping at 700 °C [58].

#### 3.1.5. σ

The σ phase, a hardening and embrittlement phase, is generally FeCr, which usually precipitates at the trigeminal grain boundary firstly and is followed by the grain boundary and it degrades not only the ductility and toughness but also the corrosion resistance due to the reduction of the amount of chromium of the austenitic matrix [78]. The precipitate temperature is generally 650–1000 °C, and it was found that it is positively related to the alloying degree of austenitic steels [37]. During aging at a high temperature for a long time, σ will appear at the non-coherent twin grain boundary. Besides the local segregation of chromium in the matrix, σ is also affected by the stability of the γ matrix. The formation of σ is mainly due to the instability of the γ matrix, resulting from the depletion of nickel for the B2-NiAl formation. σ tends to appear adjacently to the B2-NiAl [19].

In addition to the precipitates, the microstructure of AFA steels still has an austenitic matrix, grain boundaries (GBs) and sometimes appears as a L12 precipitate free zone (PFZ) in long-term service at high temperatures. On the whole, after solution treatment, the matrix of C-containing AFA steels at room temperature is a single austenitic matrix, and sometimes there is a small amount of primary MC, as shown in Figure 5a. GBs, which separate two regions of the same crystallographic structure but of different orientations, are one of the most important microstructural elements. According to the grain orientation of adjacent grains, GBs have low angle grain boundaries (LAGBs) and high angle grain boundaries (HAGBs). Hu et al. [53] claimed that HAGBs could effectively increase the energy required to activate the slip between adjacent grains, further strengthening the grain boundaries. Although GBs may be strengthened by Laves phase and B2, it was found that GBs are still weak in the microstructure [74,75].

The PFZ generally appears along the GBs, which reduces the strength of the GBs, as shown in Figure 5b. Geneva Trotter et al. [78], Andrew Peterson [75,79] and B. Hu [80] all found that the PFZ appearing in Fe-20Cr-30Ni-2Nb-5Al (at. %) AFA steel and Fe-32Ni-14Cr-3Nb-3Al-2Ti (wt.%), it provided a path for crack propagation and reduced the ductility of the material. In Fe–20Cr–30Ni–2Nb–5Al (at. %) alloy, the formation of PFZ is due to the large amount of B2 precipitating at the grain boundary, which resulted in the excessive consumption of nickel and aluminum, and γ′ near the grain boundary dissolves to supplement nickel and aluminum [74]. Therefore, the formation of PFZ generally occurs after a long creep time (more than 2000 h) [79]. The PFZ runs the length of the GBs but varies in width. Recently, Hongyuan Wen et al. [58] found that the width of PFZ was directly proportional to the square root of creep time in AFA steels containing 20 wt.% nickel. In the future, it is necessary to study further how to avoid PFZ to improve the high-temperature properties of AFA steels.

### 3.2. Factors Affecting Precipitates in AFA Steels

#### 3.2.1. Alloy Elements

It was found that alloying elements have an important influence on the precipitates, and the influencing elements of different precipitates are summarized in Table 2. The research details are described as follows.

(1)Elements affecting the carbides

Carbides-effecting elements in AFA alloys are niobium, titanium, vanadium, thallium, manganese and carbon. The work of Y. Yamamoto et al. [21] and Hu. B [81] found that higher niobium increased the volume fraction of NbC (II); however, too much niobium not only cannot increase the volume fraction of NbC (II) but would increase the amount of primary NbC (NbC (I)).

In addition, the effect of niobium on precipitates still needs to consider the ratio of Nb/C. Zhou et al. [37] and G. Muralidharan et al. [82] discovered that a less ratio of Nb/C could promote the precipitate of NbC (II). The secondary NbC (II) is the dominant precipitate when the Nb/C ratio is lower than 15 [37]. All of the above arguments suggest that the Nb/C ratio plays an important role in the formation of the secondary NbC.

Furthermore, Wen et al. [19] found a good combination of creep strength and oxidation resistance through co-alloying with Nb/Ta/V and controlling the M/C (M = Nb, Ta, V) ratio within 1.0–2.0 (in molar ratio) simultaneously.

In addition, the acceleration of tungsten on the precipitate of secondary NbC has also been reported in the literature [53].

M_23_C_6_ type carbide is easily formed when carbon exists, which tends to be coarsened or dissolved at elevated temperatures, resulting in poor corrosion resistance. It was found that low manganese and high carbon helped precipitate a high-volume fraction of M_23_C_6_ in AFA steels (12 wt.% Ni) [60].

Phosphorus is a detrimental element to the property of AFA alloys. It was found that phosphorus promoted M_23_C_6_ and inhibited the secondary NbC in the early aging stage [83].

(2)Elements affecting B2-NiAl

In AFA alloys, aluminum has a great influence on B2. It was found that when the aluminum increases from 3 wt.% to 4 wt.%, B2-NiAl would increase from 5 vol% to 10 vol% [21]. In addition to component elements, copper and niobium also have important effects on the precipitate of B2. It was found that the high content of niobium helps to increase the volume fraction of B2-NiAl [16]. In addition, copper relatively enhances the stability of nanoscale B2-NiAl due to occupying iron and nickel atoms locations and reducing the chemical driving force of the nanoscale B2-NiAl growth [84]. Zhou et al. [85] further performed the corresponding first-principle calculation and proved that copper can enhance the interaction of Ni-Al pairs and increase the precipitate of nanoscale B2-NiAl. In Fe-25Ni-18Cr-3.0Al-0.8Nb base AFA alloy, Xiangqi Xu et al. [86] found that more than 0.3 wt.% silicon stimulated B2-NiAl precipitate and reduced the Al concentration in the austenite matrix, which led to the degradation of oxidation resistance.

(3)Elements affecting the Laves phase

The study found that the Laves-Fe_2_Nb phase is dominant for the Nb/C ratio greater than 30, while both the secondary NbC and the Laves-Fe_2_Nb can co-exist if the Nb/C ratio is greater than 15. Furthermore, Lu et al. also concluded that Laves-Fe_2_Nb and the secondary NbC had a separate competitive relationship; in other words, a lower Nb/C (7.7–10) ratio is conducive to the emergence of NbC (II), while a higher Nb/C (greater than 23) is beneficial for the formation of Laves-Fe_2_Nb in the AFA alloys [37]. In AFA-W steels, the larger amount of dispersed Fe_2_W type Laves phase interrupted δ ferrite, thus avoiding the formation of harmful ferrite network and exhibited higher room temperature tensile properties [35]. Meanwhile, Jang et al. [87] and Wen et al. [55] also found that the addition of tungsten promoted the precipitate of Laves phase and reduced its coarsening rate, and inhibited the precipitate of harmful phase σ. Furthermore, from the perspective of thermodynamics, Jang et al. revealed that the addition of tungsten decreased the enthalpy of formation, solubility and diffusivity of Laves phase in AFA steels, which in turn led to an increase in volume fraction and a decrease in size [27]. In addition, the promotion of tungsten elements on the precipitate of NbC (II) has also been reported in the literature [53].

In addition to niobium and tungsten elements, Y. Yamamoto et al. found that silicon could promote the nucleation of Laves phase and stabilize the size and distribution of Fe_2_Nb phase [38], which is consistent with the research results of Yujiao Jiang et al. [88].

(4)Elements affecting γ′

γ′ phase consists of the same nickel and aluminum elements as B2. Therefore, the contents of nickel and aluminum are the main influencing factors. The study found that the addition of aluminum promoted the formation of fine particles of γ′-Ni_3_Al [38], and a similar result was found in the literature [89]. Still, in DAFA 29, aluminum addition is less effective to the precipitate of γ′- Ni_3_Al [40]. It was found that in AFA-W steel, Ni-Cu-Al L12 ordered phase was finer than in AFA steels, and the result shows that tungsten is beneficial to the precipitates in AFA steels [35]. Compared to AFA steels, AFA-W steels exhibited higher room temperature tensile properties at the same aging time, which may be caused by the strengthening effect of smaller sizes of Ni-Al-Cu phases in the grain interiors, as shown in Figure 6. In addition, the effect of zirconium on γ′ is different in different alloys. Yamamoto et al. [38,40] found that 0.3 wt.% zirconium could greatly improve the stability of the γ′ phase below 900 °C in Fe-20Cr-30Ni-2Nb (at. %) AFA alloy. While Joonoh Moon et al. [90] discovered that adding 0.2 wt.% zirconium in AFA steel (20 wt.% Ni) would lead to coarsen precipitates. Furthermore, copper also has a positive effect on γ′. The study found that between 500 °C and 700 °C, the solubility of copper in Ni_3_Al is as high as 2.8% without changing the L12 structure [91]. The research in the literature [25,26,35,92] all indicated that adding 2.8 wt.% copper to AFA steels promoted the formation of coherent L12 ordered Ni-Cu-Al phase.

(5)Elements Affecting σ

In AFA alloys, the formation of σ is related to the content of chromium. In different AFA alloys, the limit value of chromium to avoid σ decreases with decreasing temperature, and free σ can be obtained by using proper alloying additions [62]. The precipitate of σ can be avoided by reducing molybdenum, zirconium and silicon contents. Wen et al. and Jang et al. found that excessive molybdenum promoted the formation of the σ phase [55,86,93], and the addition of titanium also accelerated the precipitate of σ in high chromium and high nickel AFA steels [55]. Moreover, Zhou et al. [37] indicated that the reduction of silicon and molybdenum content is beneficial for suppressing precipitation of the σ-phase. In the study of the modified 310 AFA steel, it was found that when zirconium content is higher than 0.52 wt.%, the precipitates appear G-Ni_16_Si_7_Zr_6_ phase and σ phase earlier [93]. In addition, the addition of strong carbides forming elements vanadium and thallium can inhibit the precipitate of the σ phase [19].

It can be seen from the above results that the influence of elements on precipitates is related to the composition of AFA steels, and the influence of different compositions is different, so it is necessary to carry out more systematic research on AFA steels with different compositions.

#### 3.2.2. Pre-Strain

Many studies have found that applying a certain pre-strain in advance can introduce stable high-density dislocations into the material, which provides many nucleation sites for the precipitates and promotes the diffusion of elements. For example, in the study of DAFA 29 steel, it was concluded that the grain size (about 100 nm) was greatly reduced after applying large cold deformation [39]. Furthermore, Trotter et al. [76], Bingbing Zhao et al. [28] and Min-Ho Jang et al. [94] all found that pre-strain could make the second phases precipitate earlier and the grains finer despite the Laves phase, B2, MC and L12 ordered phase compared with the unstrained materials. Lu et al. [51] also pointed out that Laves phase precipitated earlier at the position of dislocations in comparison with the σ phase in ferritic steel when aging below 1073 K. Furthermore, Chenchen Jiang et al. found that the reason for grain refinement by pre-strain was the formation of most of LAGBs [69]. In Figure 7b,c,e,f, for the Fe-20Cr-30Ni-2Nb-5Al (at. %) AFA alloy, it can be seen that the number of precipitates increases with the increasing of cold deformation; moreover, compared with alloys without prior cold deformation, B2-NiAl and the Laves phase precipitated earlier and often co-located.

#### 3.2.3. Heat TreatmentAging

(1)Aging

Many studies show that the effect of aging on the precipitates is very complex. The precipitates are different according to aging temperatures. Zhou et al. [36] reported that the main strengthening phase in AFA steels varies with aging temperature. Below 750 °C, NbC is the major strengthening phase, while Laves-Fe_2_Nb acts as a potential strengthening phase above 750 °C.

At present, studies have found that the amount and size of precipitates are related to aging temperature and duration. According to the research by Geneva Trotter et al. [76], under the same cold deformation, the volume fraction of Laves phase precipitated during aging at 800 °C was more than that at 700 °C, as shown in Figure 7a,g,c,h and d,i,f,j, but compared with that of the pre-strained, the precipitates were obviously coarsened at 800 °C, as shown in Figure 7d,i. In addition, Zhao et al. [61] studied the aging behavior for up to 10,000 h and found that the spherical NbC (II) had lower coarsening power than the Laves-Fe_2_Nb phase and this coarsening phenomenon was also found in the literature [29,30,88].

Besides, aging also affects the order and position of precipitates. In Fe-20Ni-14Cr-2.5Al-0.13Si-1.56Nb-2Mn-0.06C-2.23Mo AFA alloy, after 500 h of aging, the globular-shaped and needle-shaped precipitates alternately precipitated on grain boundaries [88]. Wang Man et al. [29] found that with the aging time increasing to 1000 h, Laves phase began to appear at the grain boundary and then in the grain. Chen Lingzhi et al. [43] studied the aging of Fe-18Ni-12Cr-3Al AFA steel un-containing carbon and niobium at 700 °C and found that Laves-Fe_2_Mo began to precipitate along the grain boundary after aging for 10 h, while B2-NiAl phase precipitated after aging for 1000 h. Interestingly, a work by Geneva Trotter et al. [75] discovered the alternate precipitate of Laves and B2 in Fe-20Cr-30Ni-2Nb-5Al (at. %) AFA steel during aging. The alternate precipitate of B2 and Laves phases at grain boundaries was also observed when DAFA 26 was aged at 800 °C [41]. It can be seen from the above analysis that the correct use of the aging process can effectively promote the precipitates and improve the performance of AFA steels.

(2)Annealing

There are few studies on the effect of annealing on precipitates in AFA steels. The results showed that annealing could affect the number and shape of precipitates. Hu et al. [39] found that when solution annealing at 1200 °C prior to cold rolling, the AFA steels could precipitate the Laves phase with finer size and more uniform distribution. Besides, Bingbing Zhao et al. [26] studied AFA steels containing Cu annealing at different temperatures and found that compared with the material annealing at 1080 °C, the alloy annealing at 1230 °C precipitated more nanoscale MC at 700 °C. In addition, Qiuzhi Gao et al. [95] found that abnormal austenite growth occurred in normalized and annealed materials above 1250 °C, while there were some special large precipitates at the GBs of normalized materials. Similarly, the spherical dispersed Ni_3_Al particles precipitated in AFA steel based on HK 40 ((Fe-25Cr-20Ni-0.02Al-0.38C-0.77Mn-0.75Si) were finer after solution treatment [89].

In addition, some results showed that annealing also affects the grain boundary type. Liu et al. [34] found that during annealing, a small amount of LAGBs transformed into HAGBs. Similarly, Chenchen Jiang et al. also found that many HAGBs were formed in 4Al AFA steel (Fe-11Cr-20Ni-4Al-2Nb-2.25Mo-2Mn-0.14Si-0.05Cu) after annealing compared with the LAGBs after cold rolling [96]. However, different interface types have different effects on precipitates and nucleation. According to the research of Hongyuan Wen et al. [59], the coherent twin boundary (CTB) interface is not easier to nucleate and precipitate than incoherent twin boundary (ITB) and normal grain boundary (NGB) interfaces due to the difference in interface energy and diffusion rate in high-temperature service.

## 4. Corrosion Resistance

Due to the need to balance the mechanical properties and oxidation resistance within the composition range, AFA alloys developed so far do not have “strong alumina formability” like traditional Fe-Cr-Al and nickel-based alloys. At present, the corrosion resistance of AFA steels was studied in the air (including water vapour), supercritical water reactor (SCWR) and metal dusting, etc. The developed AFA steels have strong corrosion resistance in different environments and can form a single or complex oxide scale, as shown in Figure 8.

### 4.1. Oxidation Resistance in Different Environments

#### 4.1.1. In Air

The currently developed AFA steels usually form a continuous Al_2_O_3_ protective scale in high-temperature air. Like traditional stainless steel, the oxidation resistance of AFA steels with the same composition decreases with the increase in temperature. For instance, in the literature [13], AFA 2-4 (Fe-21Ni-14Cr-2.5Al-3Nb) steel has good oxidation resistance in 800 °C air or water-containing air but shows poor oxidation resistance in 900 °C air. Furthermore, the oxidation resistance of AFA steels with the same composition in air containing water vapour is worse than that in dry air, which is mainly due to the penetration of hydrogen ions will increase the solid solubility of oxygen. For example, at 800 °C in air, AFA 2–4 steel had good oxidation resistance after exposure for 5000 h, but the oxide scale peeled off after 4800 h at 800 °C in air containing water vapor [13]. And it was found that the higher the water vapor content, the worse the oxidation resistance. N. M. Yanar et al. [23] found that OC4 steel had better oxidation resistance in 3% water vapour than in 10% at 800 °C. This explains that OC4 steel can form a continuous alumina protective scale in a 3% water vapour environment but has internal oxidation in 10% water vapour. The composition of the oxide scale on the surface is relatively complex.

Table 3 summarizes the antioxidant properties of currently developed AFA steels under tested conditions. From Table 3, the oxidation temperature is mainly concentrated at 800 °C. In this environment, AFA+ Al/C alloy is tested for the longest test time (8000 h) and the weight gain of 0.2 mg/cm^2^ and is stronger than that of 880-4 alloy (150 h) and AFA based alloy (2000 h) [16]. The poorest composition of AFA steel, without niobium, tungsten, titanium, vanadium and copper, has a better oxidation resistance than AFA-SS (20Ni-14Cr-2.5Al-2.5Mo-2Mn-0.86Nb), and obtains a mass gain of 5 mg/cm^2^ after 1000 h in dry air at 800 °C [18].

Furthermore, the oxidation resistance of the developed alloys is higher than that of standard heat-resistant steels. Literature [4] showed that the weight increase (0.24 mg/cm^2^) of HTUPS 4 alloy is much lower than that of NF709 alloy, and the weight gain of HC-2 alloy is 0.2 mg/cm^2^ after exposure for 5000 h, but the weight gain of 347 steel is 0.5 mg/cm^2^ after exposure for 300 h, as shown in Figure 9 [42]. In addition, the mass increase of HT CAFA 4 is less than 1 mg/cm^2^ after 5000 h at 800 °C with 10% steam, which is better than that of HK and HP [82]. Lately, Lingfeng Zhou et al. [24] found the chromium evaporation rates from alumina-forming austenitic (AFA) alloys were ~5 to 35 times lower than that of the Cr_2_O_3_-forming Fe-based 310 and Ni-based 625 alloys at 800 °C to 900 °C in the air with 10% H_2_O.

#### 4.1.2. In Supercritical Water (SCW)

Supercritical water reactor (SCWR) has the advantages of simplified design, small size and high thermal efficiency. However, the SCW environment (above the thermodynamic critical point, with a temperature of 374.2 °C and a pressure of 22.1 MPa) is extremely corrosive to the core and fuel cladding materials. Nie et al. [97] found that the AFA steel (Fe-20Ni-14Cr-3Al-0.6Nb-0.1Ti) had a double-layer oxide scale in SCW (500 °C, 25 MPa, 25 wppb), and lower overall weight than other austenitic alloys tested under similar condition, such as 800H, D9 and 316 stainless steels. Similarly, Sun et al. [98] also found that Fe-27Ni-15Cr-5Al-2Mo-0.4Nb AFA steel could form a double-layer oxide scale in SCW (650 °C, 25 MPa, 10 ppb), which had better oxidation resistance than modified 310S steel under the same environment.

#### 4.1.3. In Molten Sodium Sulfate

In industrial and marine gas turbines with sulfur in fuel, sulfur is prone to react with sodium chloride and produce sodium sulfate. The components serving in this environment would be vulcanized or accelerated to oxidize by sodium sulfate, resulting in “hot corrosion”. Y. F. Yan et al. [99] found that when NF709-4 AFA steel was exposed to a molten sodium sulfate environment at 900 °C with air as covering gas, a dense double-layer oxide scale (alumina in the outer layer initially formed, chromium oxide in the inner layer) was formed to prevent sulphur infiltration. It had better corrosion resistance and lowered internal vulcanization than nickel-based super-alloy K438, K417 and 316L commercial steels, as shown in Figure 10. However, further study is required regarding the oxidation resistance when SO_2_ is used as covering gas.

Furthermore, B. S. Lutz et al. [100] studied the corrosion behavior of four types of AFA steel (OC4 (25Ni-14Cr), OC8 (32Ni-19Cr), OCT (32Ni-14Cr) and OCS (35Ni-14Cr)) in two simulated hearth corrosion environments at 700 °C, and found that in these environments, the alloys need more chromium to obtain the third-element effect to promote the alumina. It indicated that the high content of nickel and chromium was beneficial to the corrosion resistance.

#### 4.1.4. In Liquid Lead

Due to the strong corrosive ability of liquid lead and lead-bismuth eutectic (LBE) above 500 °C, the dissolution of nickel is prone to occur, and the operating temperature of the reactor is limited. Jesper Ejenstam et al. [52] conducted AFA steels, 20Ni (Fe-14Cr-20Ni-2.5Al-1.6Mn-2.5Mo-0.2Si-0.9Nb-0.08C) and 14Ni (Fe-14Cr-14Ni-2.5Al-1.6Mn-2.5Mo-0.2Si-0.9Nb-0.08C) alloys, which had promising corrosion resistance in oxygen controlled liquid lead containing 10-7 wt.% oxygen at 550 °C for one year, and found that the Ni in 20Ni AFA steel dissolved by liquid lead, but 14Ni AFA alloy showed excellent corrosion resistance, which is superior to 316L and 15-15 Ti alloys.

Recently, Lingzhi Chen et al. found that for Fe-18Ni-16Cr-4Al-base AFA steel, there was more obvious lead penetration and nickel dissolution with an oxygen content of 10^−9^% than the oxygen content of 10^−6^% because in the liquid lead with oxygen content of 10^−6^%, a protective oxide scale was formed [64]. In addition, Haoran Wang et al. concluded that in a low oxygen concentration environment, Al-added high manganese AFA steel formed an alumina protective scale in molten LBE, and its corrosion performance increased with the increase of aluminum content [51]. These studies indicate that low-nickel AFA steels or Ni-free AFA steels have a promising application prospect in liquid lead or LBE solution.

#### 4.1.5. In Metal Dusting

Metal dusting is a kind of serious corrosion form. A work by Jianqiang Zhang et al. [101] on the corrosion behavior of three AFA steels showed that the corrosion process all was accorded with the corrosion mechanism of classical austenitic stainless steels, and formed heavy and unique “tentacles” carbon deposits, though research [17] showed that all these alloys were able to form protective alumina scales in dry or wet air at 800 °C. While Aurelie Rouaix-Vande Put et al. [102] uncovered that in a given gas mixture composition, at atmospheric pressure, from 550 °C to 750 °C, due to the decrease in carbon activity, the corrosion resistance of Fe-23.6Ni-15Cr-7Al-0.3Al-0.43Cu-2Mn-Mo-1.5Nb-0.05V-0.06Ti-0.3W at. % AFA alloy decreased and were higher than those with a Fe-base and Ni-base alloys under the same conditions, such as T122, 800H, 347H, HR6W and so on. It was interesting to observe that the water vapour could reduce the erosion of the AFA alloys. These tests show that H_2_O can increase oxygen’s partial pressure, reduce carbon activity and metal dust corrosion, and has the value of systematic research.

#### 4.1.6. Other

In addition, corrosion resistance of AFA steels has also been tested in other environments. R. Elger et al. studied the oxidation resistance of AFA steels in a nitrogen environment and found that AFA steels with good oxidation resistance in 1000 °C air were prone to nitride. The nitriding depth predicted by the kinetic model was deeper than the test. Furthermore, no cubic chromium nitride expected to exist on the surface was detected in the exposed samples [50,103].

The research showed that under low temperatures and short exposure times in supercritical carbon dioxide, protective and continuous Al_2_O_3_ and rich (Cr, Mn) oxide formed, and with the increase of temperature and exposure time, the Al_2_O_3_ scale peeled off and a multilayer structure mainly composed of non-protective Fe_3_O_4_, (Cr, Fe)_3_O_4_, NiFe/FeCr_2_O_4_/Cr_2_O_3_/Al_2_O_3_, FeCr_2_O_4_/Al_2_O_3_, NiFe/Cr_2_O_3_/Al_2_O_3_ formed, then fractures occurred [104].

The results above suggest that the AFA alloys offer potential environmental compatibilities in not only steam/water vapour oxidation environments but also the mixed corrosive environments as well.

### 4.2. The Influence of Alloy Elements on the Oxidation Resistance

At present, the research on the influence of alloy elements on the oxidation resistance of AFA steels is mainly concentrating on dry and wet air. The key to improving the oxidation resistance of AFA steels is to increase the diffusion rate of aluminum in austenite matrix and reduce the diffusion rate of oxygen into the steels, which has formed a continuous and single protective oxidation scale. Based on the research results, according to the roles of the elements related to the formation of a protective alumina scale can be divided into three categories, the elements to stabilize the alumina scale, the elements to improve oxidation resistance and to degrade oxidation resistance.

#### 4.2.1. The Elements to Stabilize Alumina-ScaleAluminum

(1)Aluminum

Aluminum is a necessary element to form alumina. Some results showed that the higher the aluminum content in AFA steels, the better the oxidation resistance of steels is [13,15,17,20,33,65]. This is mainly because with the increase of aluminum content, the diffusion rate of aluminum from the substrate to the interface increases, so it is easier to form dense protective alumina. For instance, in the literature [15,20], the alloy B (Fe-20Ni-12Cr-3.9Al) and 3-0.6 (Fe-20Ni-14.2Cr-3Al) have better oxidation resistance than that of alloy C (Fe-20Ni-14Cr-2.9Al) and 2-0.9 (Fe-20Ni-14Cr-2.5Al).

However, it was discovered that the aluminum content is far from a simple monotonous increase function for oxidation resistance, and too much aluminum would break the balance between oxidation resistance and creep property [4]. Moreover, it was found that the oxidation resistance of AFA steels was reduced by surface aluminizing [105]. Besides, it was concluded that the improvement of oxidation resistance was related to aluminum and the ratio of M/C. When the ratio of M/C was 1~2, better comprehensive properties could be obtained [18].

It can be seen that the influence of aluminum elements on the oxidation resistance of AFA steels is complicated, and it is not the case that more is better. Thus, studying the appropriate aluminum content for AFA steels is a more important direction.

(2)Chromium

The effect of chromium on the oxidation resistance of AFA steels is called the third element effect [16,20,101], which can reduce the critical aluminum content to form protective alumina, but too many additions of chromium can also lead to the formation of brittle σ phase [106]. Therefore, it is very important to obtain the relationship between the critical value of chromium and aluminum to form an alumina protective scale. Recently, Taymaz Jozaghi et al. [18] received the relationship figure by considering the third element effect of chromium when designing AFA steels with the idea of a genetic algorithm (Figure 11). Figure 11 shows the critical chromium content that can produce the effect of the third element in the Fe-Al-Cr alloys system so that the chromium and aluminum content can be optimized, which will undoubtedly bring great convenience to the design of the alloy composition.

(3)Niobium

Coincidentally, many studies have shown that niobium addition was found to be a key to oxidation resistance [15,62,85], and its influence on oxidation resistance is more sensitive than that of aluminum. The addition of niobium can improve the solubility of Cr in the matrix, which is beneficial to the formation of alumina due to the third element effect. In addition, as mentioned in Section 3.2, the high content of niobium is helpful to oxidation resistance due to the increased volume fraction of B2-NiAl [16]. For instance, M. P. Brady et al. [86] reduced the niobium content in HTUPS 4 (alloy 5: 2.5Al/0.95Nb) to alloy 6 (2.5Al/0.16Nb). The oxidation resistance of the alloy in the air at 800 °C is decreased with a small amount of scale spalling. In addition, in the study of cast high nickel AFA steels, it was found that the AFA steels adding niobium and carbon can form a continuous alumina protective scale in air containing 10% water vapour at 1100 °C [47].

In addition, the content of niobium to form the alumina protective scale is also related to aluminum and nickel. In [62], while increasing the nickel from 20 to 26 reduces the content of niobium and aluminum needed to form alumina. However, of late, it has been found that AFA alloys without niobium can also form an alumina protective scale at 800 °C [18].

In fact, the negative role of niobium addition in the formation of Al_2_O_3_ scales at above 1000 °C was reflected by the excellent oxidation resistance in some other AFA steels without the niobium addition. In the Fe-25Ni-10Cr-5Al-0.03C AFA steel without the niobium addition, the excellent oxidation resistance could be obtained when oxidized in air at 1100 °C, but scales spallation and mass loss were observed in the Fe-25Ni-15Cr-4Al-2.5Nb-0.09C base AFA steel when the oxidation temperature was increased to 1000 °C in the air with 10% water vapour [16]. In addition, Lin Shen et al. [107] found that due to the niobium addition resulting from the suppressed outward diffusion of aluminum atoms in the formation of the Fe_2_Nb phase, the oxidation resistance degraded.

This indicates that niobium has a complex effect on oxidation resistance, which needs further study.

#### 4.2.2. The Elements to Improve Oxidation ResistanceNickel

(1)Nickel

An unexpected finding in AFA steels is that nickel not only can stabilize austenitic structure but also has an important influence on oxidation resistance. As aluminum and chromium are ferrite stable elements, increasing aluminum requires increasing nickel content. It was found that increasing nickel content would increase the applicable environmental temperature of AFA steels. In the literature [15], AFA steel D (Fe-15Ni-12Cr-3Al) has the worst oxidation resistance than that of AFA steel C (Fe-20Ni-14Cr-3Al), and the oxidation resistance of AFA steel A (Fe-26Ni-14Cr-3Al) in the air at 900 °C increased from 500 h to over 4000 h.

In addition, the increase in nickel should be accompanied by an increase in chromium. The cast CAFA 4 containing nickel (25 wt.%) and chromium (14 wt.%) could form an alumina protective scale after being exposed to air with 10% water at 800 °C for 4000 h [82]. Furthermore, adding higher contents of nickel (35 wt.%) and chromium (25 wt.%) and active elements would increase the upper limit of forming temperature of alumina protective scale to 1100 °C [46], indicating that nickel can significantly improve oxidation resistance.

It was found that AFA steels would change from external oxidation to internal oxidation when exposed for a long time [20]. It was uncovered that niobium and nickel could increase the internal oxidation transition temperature more than aluminum alone [15,17].

(2)Carbon and boron

The study found that carbon and boron can precipitate enhanced carbides and borides, and the oxidation resistance is also improved to a certain extent. Xu X et al. [16] found if the content of carbon increased from 0.1 to 0.2 (wt.%), or the content of boron from 0.01 to 0.1 (wt.%), the internal oxidation temperature of AFA steel in water vapour environment was increased by 150 °C, decreased detrimental effects on the internal/external oxidation transition [16]. This is because carbon and boron promote the precipitate of Ni_3_Al phase, which can provide the location of hydrogen segregation in water vapour. It is consistent with the results of the research on the OC4 AFA alloy [40].

(3)Hafnium and yttrium

Since the active elements hafnium and yttrium can effectively absorb hydrogen in water vapour, it can improve the antioxidant capacity of AFA steels. In literature [20], it was found that the 4-1HfY alloy, which was added 0.15% wt.% hafnium and 0.02% wt.% yttrium, did not suddenly lose oxidation resistance like the alloy 4–1 (25Ni, 4Al), but showed a gradual mass loss, which significantly improved the oxidation resistance at 800 °C. In addition, the research on the oxidation resistance in 1200 °C steam also found that the addition of yttrium could reduce the growth rate of alumina and inhibit the peeling of the oxide scale [108]. In general, to optimize the alumina scale (slow growth rate, increased scale adherence), Hf/C and Y/S should be >1 [109].

#### 4.2.3. The Elements to Degrade Oxidation Resistance

Titanium and vanadium are high-temperature strengthening elements and are often added to traditional stainless steels, but it was found that adding titanium and vanadium at the same time could reduce the oxidation resistance of AFA steels [4]. Surprisingly, however, adding titanium or vanadium alone and controlling the addition amount within 0.5 wt.% had little effect on oxidation resistance [17]. This was consistent with the study of the AFA steels based on 310S by D. H. Wen et al. [19], and it was found that the better oxidation resistance was obtained by mixing niobium and (titanium, vanadium and thallium) into AFA steels.

#### 4.2.4. Other

The effect of silicon, manganese and tungsten on oxidation resistance was also studied. Because manganese has a stronger affinity with oxygen than aluminum, in Fe-20Ni-14Cr-2.5Al-0.8Nb and Fe-25Ni-18Cr-3Al base AFA steels, Mn-rich oxides can easily form under an oxidizing atmosphere, which destroyed the continuous alumina protective scale and deteriorated the oxidation resistance. This result was also found by Brady et al. [42].

The effect of silicon on oxidation resistance is related to its content. Xiangqi Xu et al. [86] discovered that for NF709 AFA steel, in 800 °C water vapour air, when silicon was less than 0.5, the oxidation resistance of steels could be improved, but when it exceeded 0.5 wt.%, the oxidation resistance dropped sharply. In addition, for 22Cr-25Ni high chromium AFA steels, Nan Dong et al. [110] found that through the first principles analysis, silicon could improve the oxidation resistance by increasing the adhesion of interface among the multi-layer oxide scale (Fe/Al_2_O_3_/Cr_2_O_3_). The enhancement of tungsten on oxidation resistance mainly lies in the limitation of W-rich secondary phase to oxygen diffusion precipitated from the metal and oxygen interface [111].

Except for alloy composition, the heat treatment process also has a certain effect on the oxidation resistance of AFA steels. It was found from the literature [20] that the pre-aged reduced the oxidation resistance of AFA steels in water-containing air because B2 phase did not precipitate directly under the alumina protective scale. On the contrary, the formation of B2 seriously affected the oxidation resistance because of consuming aluminum elements. In addition, it was found that the introduction of dislocations provided a channel for aluminum diffusion to the substrate surface and a nuclear site for the formation of B2, which was beneficial to the oxidation resistance of AFA steels [112].

## 5. Mechanical Properties

At present, the research on the mechanical properties of AFA steels mainly focuses on short-time mechanical properties and high-temperature creep properties, while the hot working of AFA steels started relatively late, with few studies. It was found that the mechanical properties of AFA steels are closely related to its microstructure and the type, size and distribution of precipitates at high temperature, and its strengthening mechanism is mainly precipitate strengthening and dispersion strengthening. This paper comprehensively analyzes the hot workability, short-term mechanical properties, and creep properties of AFA steel.

### 5.1. Hot Work-Ability

The research on the hot workability of AFA steels mainly focuses on hot compression simulation experiments. Currently, the hot working performance of AFA steels is mainly tested on a Gleeble testing machine, with the temperature ranging from 700 °C to 1200 °C, constant strain loading and a strain rate generally ranging from 0.005 s^−1^ to 5.0 s^−1^. According to the measured data, a reasonable hot working range of alloys can be obtained.

S. Sun et al. [31] studied the AFA steel, Fe-16Cr-3Al-2W-0.3Si-0.4Nb-0.04Y, and found that at a strain of 0.1, the optimal hot working condition occurred in the temperatures range of 1050–1075 °C and strain rates range of 0.03 s^−1^-0.3 s^−1^. Rui Luo et al. [106], Min-Ho Jang et al. [53], and Qiuzhi Gao et al. [113] carried out thermal simulation tests on the hot workability of the studied AFA steel and obtained the suitable temperatures and strain rate for hot working. From the current research situation, the thermal simulation research is insufficient, such as the influence of processing passes and processing methods on the performance is still blank. In addition, the actual verification is still very lacking, which has a long way to go with the popularization and use of materials.

### 5.2. Short-Time Mechanical Properties

At present, the short-time mechanical properties of AFA steels are mainly tested in yield and ultimate tensile strength (YS and UTS), respectively. Strength tests generally include high temperature and room temperature (RT), and the temperature ranges from 600 °C to 850 °C. At a high temperature, it is generally lower than that at RT. For example, Y. Yamamoto et al. [21] found that the YS and UTS of A-0.9 steel were 523/650 MPa at RT and 349/407 MPa at 750 °C. Similarly, the YS and UTS of AFA steel with a nickel content of 20–25 wt.% and 12 wt.% all decreased with the increase in temperature [60].

#### 5.2.1. Reinforcing Phase

The high-temperature mechanical properties of AFA steels are typically improved by precipitate strengthening. The main hardening precipitates in these alloys are MC (M stands for metal like niobium and titanium) carbide and intermetallic compounds such as B2-NiAl, Fe_2_(Mo, Nb)-Laves and Ni_3_Al phases [22,37,39,41,52,68]. It was found that MC and/or M_23_C_6_ are the main reinforcing phases of medium nickel (20–25 wt.%) and low nickel (12 wt.%) AFA steels, while L12 (γ′-Ni_3_Al) is that of high nickel (32 wt.%) AFA steels. For example, the YS and UTS of the AFA steel with Fe-20Ni-14Cr-2.69Al at 700 °C were 208 MPa and 420 MPa, respectively, because nanoscale Nb(C, N) was the main reinforcing phase [90]. And NF709-4 AFA steel with Fe-25Ni-18Cr-3Al is strengthened by the precipitate of a large amount of NbC (II) at 750 °C, and the YS and UTS reach 310–335 MPa and 480–500 MPa [65]. Furthermore, Qiuzhi Gao et al. [84] found the B2-NiAl phase led to an improved comprehensive property with high tensile strength (1464 MPa) and good ductility.

#### 5.2.2. Short-Time Properties of AFA Steels Currently Tested

Uniaxial tensile property is a key material index that provides a reliable basis for further creep test. The short-term mechanical properties of AFA steels are mainly unidirectional tensile properties, which may be treated by cold deformation and aging before testing. The short-term mechanical properties of AFA steels developed at present are summarized in Table 4. It should be noted that the high-temperature YS and UTS of advanced austenitic stainless steels, such as Alloy 709, are around 200 and 400 MPa at 1023 K (750 °C), respectively [114]. Compared to the test results in Table 4, the tensile properties are comparable to those of the AFA alloys.

(1)Tensile properties after pre-strain

It can be seen from Section 3.2 that pre-strain has a great influence on precipitates, so the strength will also change after pre-strain. For instance, in the literature [21], after pre-strain, room temperature strength is greatly improved, and the YS exceeds 500 MPa (250 MPa at no-Cold Work (CW)) and remains 350 MPa (200 MPa at no-CW) at 750 °C. Similarly, Bin Hu et al. [115] found that the YS (above 1000 MPa) was two times higher than raw materials at room temperature. In addition, the strain rate jump test showed that the thermomechanical treated steels at 600 °C and 700 °C had a greater strain rate sensitivity and lower activation volume [39,115].

(2)Tensile properties after aging

Although the second phase can precipitate as aging, the strengthening effect is complex. Most studies found that the strength of the alloy increased after aging. A study on mechanical properties after aging 500 h [22] found that the YS and UTS were improved from 240–270 and 570–670 MPa to 400–450 and 890–930 MPa, respectively. This is consistent with a study in reference [21], whereby the YS was 268 MPa at room temperature, and after aging at 750 °C for 50 h, the YS reached about 400 MPa. The phenomenon of aging strengthening has also been found in other studies. Geneva Trotter et al. [75] researched the strength of aging at 800 °C and found that due to the precipitate of a large number of Laves phase and B2, the YS increased from 205 MPa to 383 MPa after aging at 1325 h. Coincidentally, the same findings were found in the studies of Wang Man et al. [29,43] and Nan Dong et al. [35]. In Figure 12, at room temperature, with the increasing aging time the UTS of Fe-18Ni-12Cr-based AFA steel after aging at 700 °C increased, but the UTS at 700 °C decreased with aging time due to the dynamic recovery softening mechanism [29].

However, the opposite occurred when the aging temperature is higher than the hard, brittle transition temperature. For example, because 750 °C exceeded the ductile–brittle transition temperature of B2, which made B2 lose its reinforcing effect and the high-temperature strength changed little [22]. On the other hand, it was found that the strength decreases due to the coarsening of precipitates with the increasing aging time. For instance, the YS of DAFA29 and DAFA26 steels decreased when aging at 800 °C due to the coarsening of L12 [81], and the change of mechanical properties with aging time is shown in Figure 13. Similarly, after isothermal aging at 973 K, Laves phase and B2 coarsened with aging time, so the YS at room temperature increased slowly and then gradually decreased gradually [30,88].

(3)Others

There are few studies on the properties and wear resistance of AFA steels in SCWR. Hongying Sun et al. [98] found that the strength of AFA steels was more sensitive to temperature than oxygen concentration, and the YS and UTS (330 and 435 MPa) in SCWR at 650 °C and 25 MPa are superior to that of modified-310 steel (160/387 MPa) and commercial HR3C (205/392 MPa). On the other hand, Y. F. Sun et al. [89] discovered that the wear resistance of AFA steels increased by 80% compared with HK 40 due to the uniform distribution of spherical nano-Ni_3_Al in the matrix.

### 5.3. Creep Performance

Some studies have found that the nanoscale carbides, B2, Laves phase, and γ′ phase can improve the creep resistance of AFA steels. While Laves phase has a relatively poor enhancement to creep resistance, if companying MC, its creep properties can be significantly improved [48]. It was found that the fine and dispersed B2-NiAl combined with other reinforcing phases plays an important role in improving the creep properties of AFA steels. In the literature [32,88], the B2-NiAl phase can enhance the high-temperature creep resistance of modified 2.5 Al-AFA steel. Furthermore, the combination of fine B2 and Laves phase increases the creep life of AFA steels at 700 °C [27,94]. In addition, the precipitate of the γ′ phase can also improve the creep performance of AFA steels [25,92]. Yamamoto et al. [12] found that nano-sized Laves-Fe_2_Nb and dispersed Ni_3_Al, particularly around 100 nm in diameter, were effective in improving the creep resistance.

#### 5.3.1. Research Methods of Creep

The data regarding the creep rupture strength of materials can be obtained by a high-temperature creep test. Still, the design life of many high-temperature parts is as long as 100,000 h, so it is obviously unrealistic to obtain the complete creep data within the design life of materials. Therefore, accelerated creep tests such as the stress and temperature acceleration methods are usually used to obtain enough rupture strength data. Based on these data, the creep life of materials can be predicted by the isotherm method [116], Larson–Miller parameter method [80], Manson–Haferd parameter method [117] and Monkman–Grant relationship [118].

In the study of the creep life of AFA steels, Y. Yamamoto et al. [4,21,40,60,81] and G. Muralidharan et al. [46,82] applied Larson–Miller parameter method. Wang Man et al. [119] inferred the creep rupture time of Fe-18Ni-12Cr-based AFA steel under 700 °C, 80 MPa by Monkman–Grant relationship, as shown in Figure 14. The related creep mechanism was found to be dislocation creep. It was found that B2-NiAl contributed to improving the material’s creep resistance. Creep failure was a result of microstructural degradation of precipitates coarsening. It can be seen that the study of high-temperature creep of AFA steels still needs to be further deepened, especially in the selection and accuracy of creep life analysis methods.

#### 5.3.2. Creep Properties of AFA Steels

At present, the creep property of AFA steels is comparable to those of traditional heat-resistance steels under the same condition. The creep properties of AFA steels are summarized in Table 5 HTUPS 4, which was firstly developed by the ORNL team, has a creep life of 2200 h at 750 °C/100 MPa, which can be comparable to NF709 (2000–6000 h) and 347 stainless steel (less than 300 h) [4], and the creep properties of 20 wt.% nickel high niobium AFA steels can also be comparable to NF709 [51]. In addition, the creep life of AFA steel with L12 as reinforcement phase at 750 °C/100 MPa is 5282 h, which is 20 times that of A286 under the same condition [40,79]. The creep life of the developed low Ni AFA steel at 650 °C/250 MPa is 2.5 times that of super 304H [60]. The creep fracture life of the developed cast AFA alloys is higher than that of the HK type alloy, which reached more than 10,000 h under the test condition of 750 °C/100 MPa [46,82]. Recently, the AFA steel with simple composition without the addition of niobium and carbon developed by Wang Man et al. had a creep life of 1038 h at 700 °C/100 MPa [119]. It can be seen that the creep life of AFA steels developed at present is comparable to that of existing heat-resistant steels.

## 6. Prospects

In this paper, the research status of the composition design, corrosion resistance, mechanical properties and high temperature creep properties of AFA steels since 2007 are summarized. Based on the corrosion resistance and creep resistance of AFA steels in high-temperature environments, it has broad application prospects in terms of thermal power, nuclear power and other fields (reactor cladding, super-heater and heat exchanger pipes). The author’s research group currently carried out research on the uniform corrosion and stress corrosion of AFA heat-resistant steels in supercritical water [98]. And based on the research on the preparation and aging stability of AFA heat-resistant steel, authors studied the creep properties of Fe-18Ni-12Cr-3Al AFA heat-resistant steels [29,119]. In addition, by fitting and analyzing the creep curves of Fe-18Ni-12Cr-3Al and Fe-27Ni-15Cr-5Al AFA heat-resistant steels at 700 °C, it was found that the latter has better creep properties. For future research directions, we have the following suggestions.

(1)AFA steel composition is further optimized. The better the composition optimization effects of AFA steels, the stricter the composition control can be, and the cost will be reduced as much as possible to promote the industrialization of AFA steels.(2)Currently developed AFA steels easily form alumina protective scales in air. In contrast, the protective scales in other harsh environments are more complex, and the research results are relatively few, so systematic study is still needed.(3)Alloying elements have an important influence on the formation of precipitates and oxidation resistance, especially nickel, chromium, niobium, aluminum, etc., but the influence of aluminum on creep needs to study systematically. At present, our research group is carrying out research in this area. Due to the time-consuming creep test, there is no specific research result.(4)At present, there is lagging research on the forming performance of AFA steels. The forming performance of the materials puts forward a better optimization plan for the design of the alloy composition and structure. Nowadays, the research on the forming performance of AFA steels is only limited to the hot compression simulation test, and the hot forming and cold forming of AFA steels are still not available. Corresponding processes such as the extrusion of heat transfer pipes and welding, etc., must be accelerated to improve the performance and applications of AFA steel.

## Figures and Tables

**Figure 1 materials-15-03515-f001:**
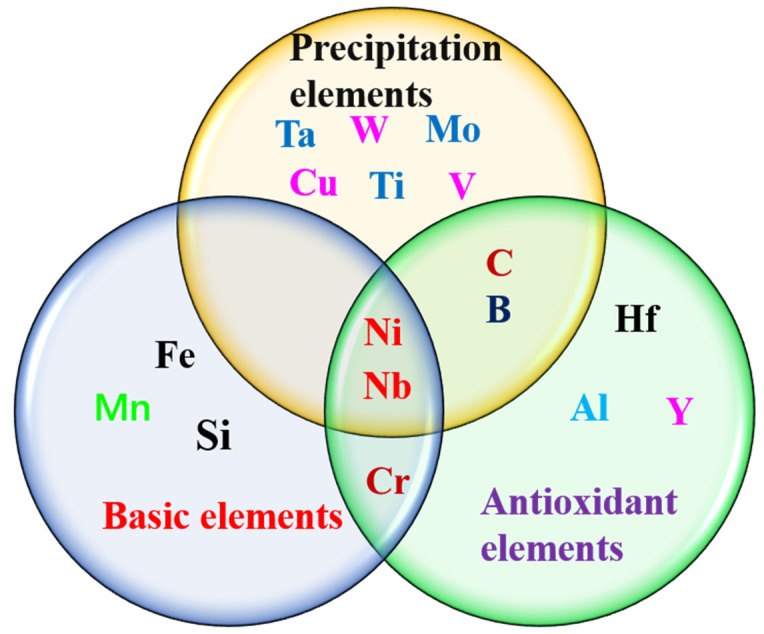
The alloy elements in AFA steel.

**Figure 2 materials-15-03515-f002:**
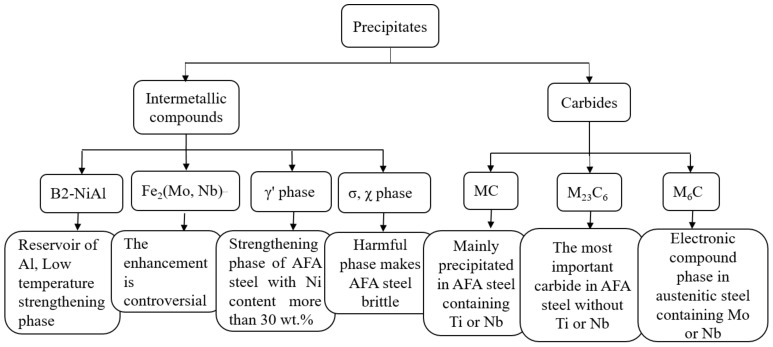
Precipitates in AFA steel at high temperatures.

**Figure 3 materials-15-03515-f003:**
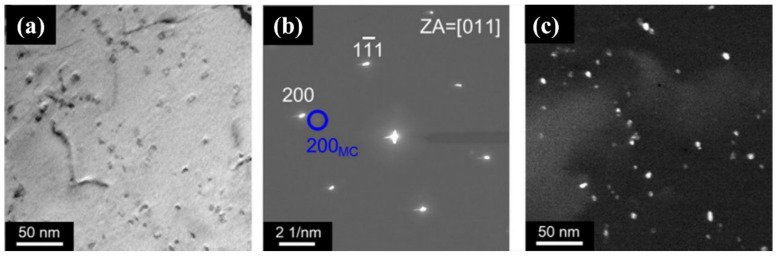
TEM images of AFA steel (**a**–**c**) after tensile testing at 700 °C/strain rates of 8 × 10^−5^ s^−1^: (**a**) are bright-field images; (**b**) diffraction patterns; (**c**) dark-field images of MC precipitates.

**Figure 4 materials-15-03515-f004:**
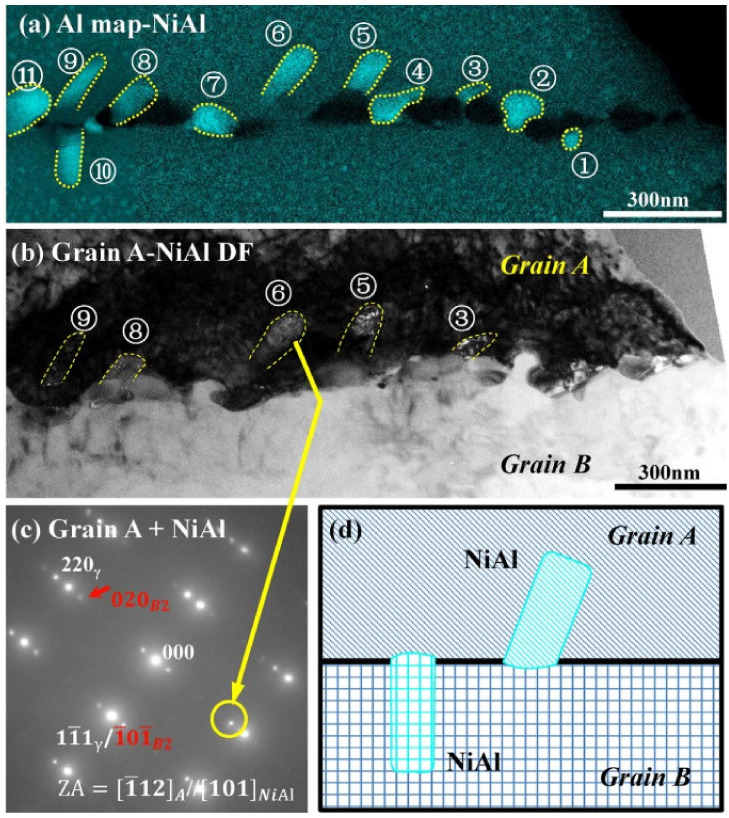
Preferred growth orientation of the NiAl phase. (**a**) STEM-EDS mapping of Al element; (**b**) TEM DF image using the NiAl diffraction spot in (**c**); (**c**) the corresponding SAD pattern of grain A with ${\;[\bar{1}12]}_{\gamma }{//[101]}_{NiAl}$; (**d**) schematic view of NiAl location at GB [63].

**Figure 5 materials-15-03515-f005:**
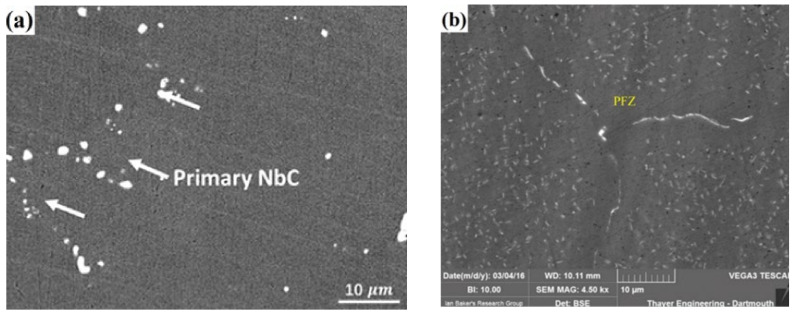
(**a**) BSE images showing the microstructure of solution treatment AFA-1 [61]; (**b**) BSE images of Fe-20Cr-30Ni-2Nb-5Al solution treatment at 1250 °C for 24 h and annealed for 24 h at 800 °C after creep testing for 500 h [69].

**Figure 6 materials-15-03515-f006:**
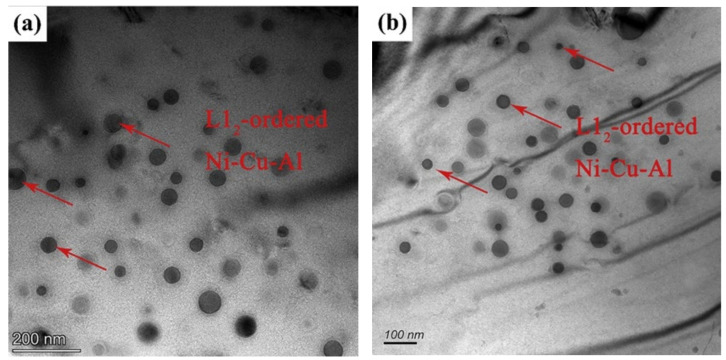
TEM micrograph of nanoscale L1_2_-ordered Ni-Cu-Al particles in (**a**) AFA and (**b**) AFA-W after aging at 700 °C for 1000 h.

**Figure 7 materials-15-03515-f007:**
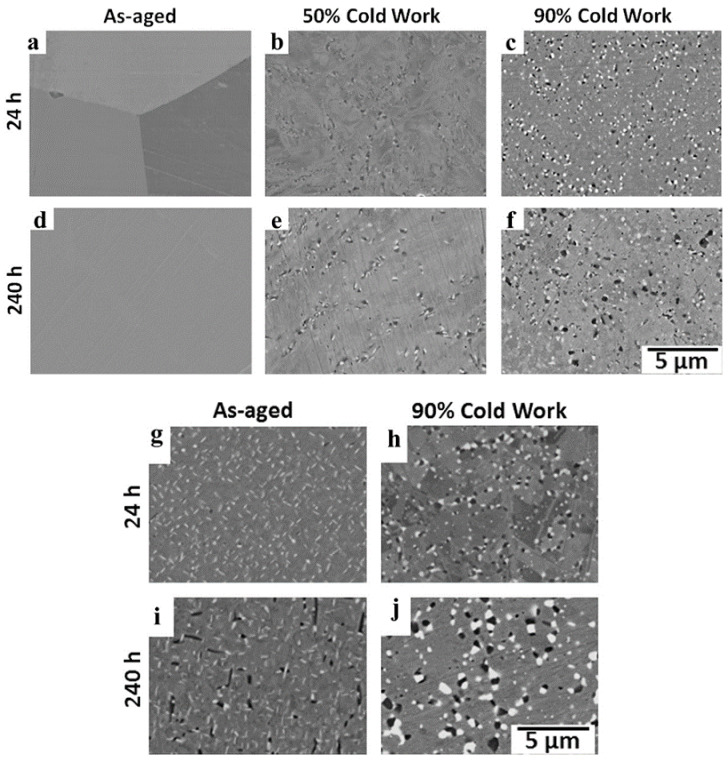
Backscattered electron images from specimens: (**a**) 24 h aged at 700 °C; (**b**) 24 h after 50% cold work aged at 700 °C; (**c**) 24 h after 90% cold work aged at 700 °C; (**d**) 240 h aged at 700 °C; (**e**) 240 h after 50% cold work aged at 700 °C; (**f**) 240 h after 90% cold work aged at 700 °C; (**g**) 24 h aged at 800 °C; (**h**) 24 h after 90% cold work aged at 800 °C; (**i**) 240 h aged at 800 °C; and (**j**) 240 h after 90% cold work aged at 800 °C [76].

**Figure 8 materials-15-03515-f008:**
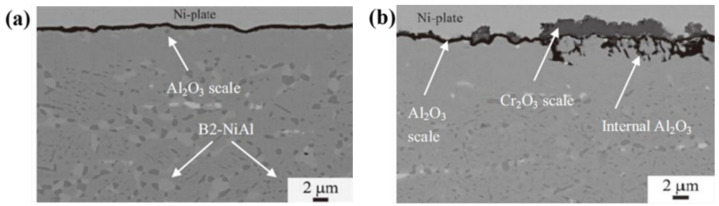
Oxidation scale of AFA steels: (**a**) single thin scale; (**b**) complex oxide scale [86].

**Figure 9 materials-15-03515-f009:**
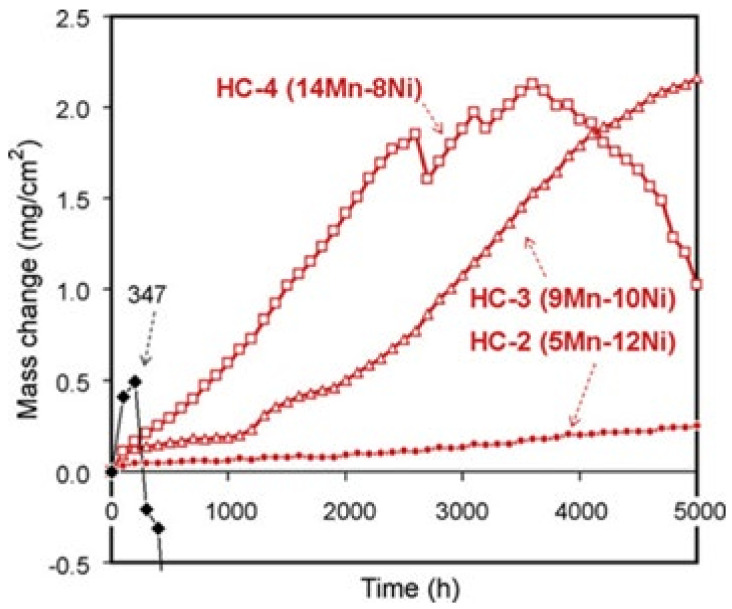
Cyclic oxidation data of 14Cr-2.5Al alloys at 923K in air +10% water vapour [42].

**Figure 10 materials-15-03515-f010:**
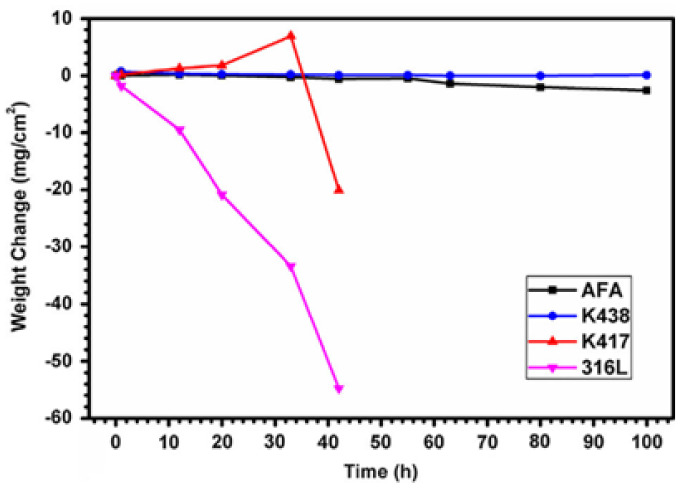
Weight change of the specimens hot-corroded at 1173 K in molten sodium sulphate [99].

**Figure 11 materials-15-03515-f011:**
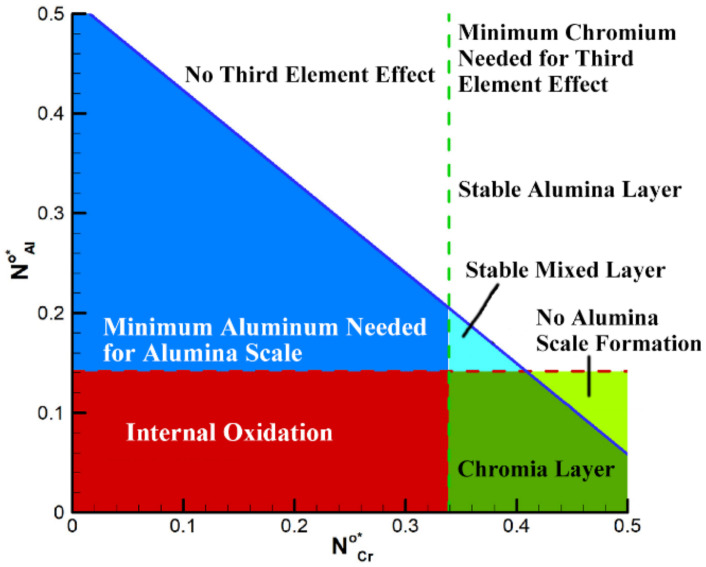
Interpretive visualization of the third element phenomenon in the Fe-Al-Cr alloy system, predicting the type of oxide scale formed as a function of Al and Cr content [18].

**Figure 12 materials-15-03515-f012:**
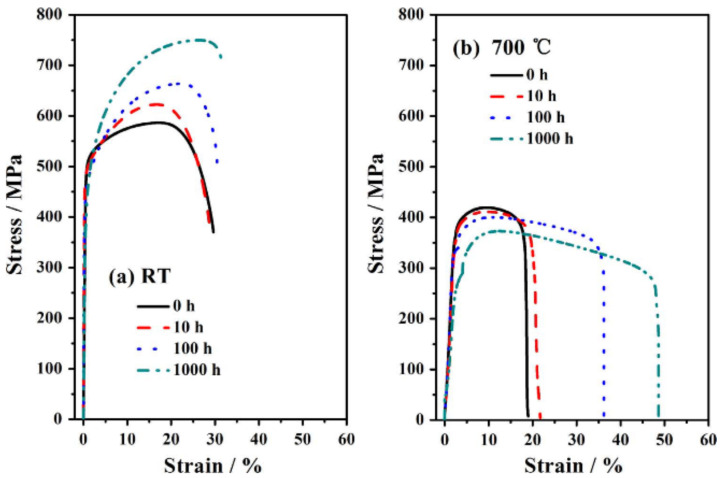
(**a**) Stress–strain curves at RT and (**b**) 700 °C [29].

**Figure 13 materials-15-03515-f013:**
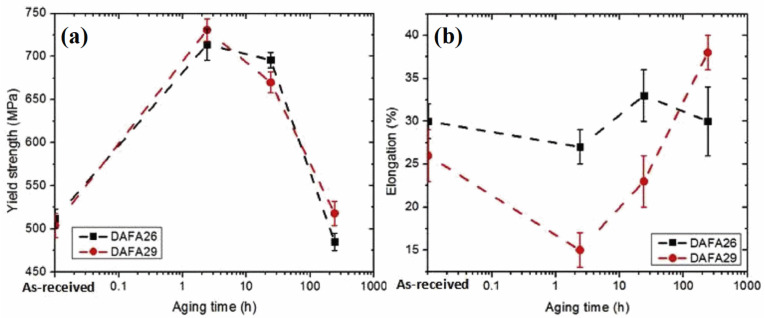
(**a**) Yield strength and (**b**) elongation of DAFA26 and DAFA29 as a function of aging time at 800 °C [81].

**Figure 14 materials-15-03515-f014:**
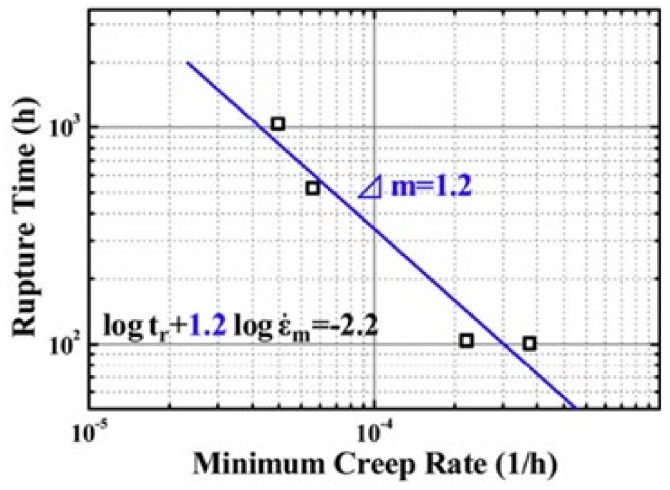
Relationships between creep rupture time and the minimum creep rate of Fe-18Ni-12Cr-based AFA steel at 700 °C [119].

**Table 1 materials-15-03515-t001:** Typical AFA alloys developed.

Name	Composition (wt.%)	References
**Baseline AFA: Fe-(20–25) Ni-(14–15) Cr-(2.5–3.5) Al-(1–3) Nb-2Mn-(0–4) Mo/W-0.5Cu + B, C, P wt.%** **Corrosion limit: Up to ~700–850 °C**		
AFA 2-1 (HTUPS 4)	Fe-20Ni-14.3Cr-2.5Al-0.9Nb-2.5Mo-2Mn-0.15Si-0.08C-0.01B-0.04P	[4]
A	Fe-26Ni-14Cr-2.8Al-0.6Nb-1.3Mo-0.15W-0.2Mn-0.2Si-0.04C	[15]
AFA + Al/C	Fe-25Ni-15Cr-4Al-2.5Nb-0.1C-0.01B	[16]
13	Fe-21Ni-14Cr-2.3Al-3Nb-0.19V-0.02C-0.01B	[17]
AFA-SS-M	Fe-17Ni-15Cr-3Al-2Mo-9Mn-0.2Si	[18]
AFA-NbTa	Fe-20Ni-18Cr-2.5Al-2.3Mo-0.45Nb-0.89Ta	[19]
3–0.6	Fe-20Ni-14Cr-3Al-0.6Nb-2Mn-2Mo-W-0.1Ti-0.5Cu-0.13Si-0.1C-0.04 P-0.001S	[20]
3–0.4	Fe-20Ni-14Cr-3Al-0.4Nb-2Mn-2Mo-W-0.1Ti-0.5Cu-0.13Si-0.1C-0.04 P-0.007S	
3–2.5	Fe-20Ni-14Cr-3Al-2.5Nb-2Mn-2Mo-0.9W-0.1Ti-0.5Cu-0.15Si-0.08C-0.04 P	
A-0.9	Fe-14Cr-2.5Al-20Ni-0.9Nb-0.01Cu-2.5Mo-2Mn-0.15Si-0.02V-0.01Ti-0.075C-0.01B-0.043P	[21]
B-1.0	Fe-12Cr-3Al-20Ni-Nb-0.5Cu-2Mo-2Mn-0.15Si-W-0.1C-0.007B-0.002P	
C-1.0	Fe-14Cr-4Al-20Ni-Nb-0.5Cu-2Mo-2Mn-0.15Si-W-0.1C-0.007B-0.002P	
AFA1	Fe-14Cr-3Al-20Ni-Nb-0.5Cu-2Mo-2Mn-0.15Si-W-0.1C-0.007B-0.02P	[22]
AFA5	Fe-12Cr-4Al-20Ni-Nb-0.5Cu-2Mo-2Mn-0.15Si-W-0.1C-0.007B-0.02P	
AFA6	Fe-12Cr-4Al-25Ni-Nb-0.5Cu-2Mo-2Mn-0.15Si-W-0.1C-0.007B-0.02P	
OC 4	Fe-25Ni-14Cr-3.5Al-2.5Nb-2Mn-0.5Cu-2Mo-0.16W-0.16Si-0.1C-0.05Ti-0.05V-0.02P-0.009B-W	[23,24]
AFA-Cu	Fe-20Ni-14Cr-2.5Mo-2.8Cu-2.25Al-2Mn-0.5Nb-0.2V-0.15Si-0.04C-0.01B	[25,26]
AFAW	Fe-20Ni-14Cr-2.5Al-Nb-0.16Si-0.1C-0.02V-2W	[27]
A-0	Fe-20Ni-14Cr-1.9Al-0.5Nb-0.18Si-0.046C-2.36Mo-1.26Mn-0.22V-0.012B	[25]
AFA-Cu	Fe-20Ni-14Cr-1.9Al-0.5Nb-0.18Si-0.046C-2.36Mo-1.26Mn-0.22V-0.012B-2.8Cu	[25,28]
Fe-18Ni-12Cr based	Fe-20Ni-12Cr-2Mo-0.4Si-0.03Mn-2.3Al-0.8Nb-0.02C	[29]
316 based AFA	Fe-18Ni-16Cr-2Mo-0.3Si-0.03Mn-4Al-0.4Nb-0.01C	[30]
AFA	Fe-25Ni-16Cr-3Al-2W-0.3Si-0.4Nb-0.04Y	[31]
2.5Al AFA	Fe-14Cr-20Ni-2.5Al-1.5Nb-0.13Si-0.01Ti-2.2Mn-2.2Mo-0.06C-0.03W	[32]
22Cr-25Ni AFA	Fe-25Ni-22Cr-0.45Nb-1.5/2.5/3.5Al-0.8Mn-2.75Cu-0.2Si-0.06C	[33]
310S AFA	Fe-(18-23)Cr-20Ni-2.5Al-2.3Mo-0.08C-0.45Nb-Ti/V/Ta	[19]
AFA	Fe-14Cr-20Ni-2.5Al-1.5Nb-0.13Si-0.01Ti-2Mn-0.06C-2Mo-0.03W	[34]
Fe-15Cr-25Ni-3Al-NbWCu	Fe-15Cr-25Ni-3Al-0.5Nb-0/2.5W-2.8Cu-0.01P-0.8Mn-0.3Si-0.08C	[35]
**High performance AFA: Fe-(25–30) Ni-(12–15) Cr-(3.5–4.5) Al-(1–3) Nb-0.1Hf/Zr-0.02Y-2Mn-(0–4) Mo/W-0.5Cu + B, C, P** **Corrosion limit: up to 800–950 °C**		
4–1(Hf, Y)	Fe-25Ni-12Cr-4Al-Nb-2Mn-2Mo-W-0.5Cu-0.15Si-0.1C-0.025P/(−0.14Hf-0.024Y)	[20]
AFA	Fe-25Ni-18Cr-3Al-1.5Nb-1.5Mo-0.15Si-0.08C-0.01B-0.04P-0.15Hf-0.01Y	[36]
AFA + Al/C	Fe-25Ni-15Cr-3Al-2.5Nb-0.1C-0.01B-0.009Y-0.13Hf +Cr, Si, Al, C, B	[16]
NF709 base AFA	Fe-25Ni-18Cr-3Al-(0.5-1.5)Nb-1.5Mo-0.15Si-0.08C-0.01B-0.04P-0.15Hf-(0–0. 1)Y/(−0.1Ti)	[37]
OC11	Fe-25Ni-15Cr-4Al-2.5Nb-2Mn-0.15Si-2Mo-0.11C-0.01B-0.5Cu-0.18Hf-0.03Y	[24]
**AFA-super alloy: Fe-(30–35) Ni-(14–19) Cr-(2.5–3.5) Al-3Nb + B, C, Ti, Zr** **Corrosion limit: up to 750–850 °C**		
OC 8	Fe-32Ni-18Cr-3Al-3.3Nb-2Mn-0.15Cu-0.15Mo-0.16W-0.13Si-0.1C-0.05Ti-0.05V-0.14W	[23]
2–3.3	Fe-32Ni-19Cr-2.4Al-3.3Nb-2Mn-0.01C-0.005 P	[20]
DAFA 29 (32ZCB)	Fe-20Cr-30Ni-2Nb -5Al (at. %) Fe-32Ni-14Cr-(3–4)Al-3Nb-(0–0.3)Zr-0.1C-0.01B-0.15Si-(1–3)Ti	[38,39,40]
DAFA 26(32Z)	Fe-14Cr-32Ni-3Nb-0.15Si-3Al-2Ti-0.3Zr-0.1C-0.01B-0.1Mo	[40,41]
**low nickel (wt_Ni_. % < 20%), Corrosion limit: Up to ~650 °C, Lower-cost, low-Ni AFA alloy**		
HC-2	Fe-14Cr-5Mn-12Ni-3Cu-2.5Al-0.6Nb-0.1C-0.007B-0.002N	[42]
Simple AFA 18-12-Al 18-12-AlNb	Fe-18Ni-12Cr-2Mo-0.3Si-3Al Fe-18Ni-12Cr-2Mo-0.3Si-2.5Al-0.6Nb-0.02C	[43,44]
**Cast AFA: Fe-25Ni-14Cr-(0–4) Mo/W-(0.5–1) Si-3.5Al-2Mn-1Nb-0.05V-W-0.05Ti-(0.2–0.5) C-0.5Cu + B** **Corrosion limit: up to ~750–850 °C**		
CAFA 4	Fe-25Ni-14Cr-2Mo-0.5Si-3.5Al-2Mn-1Nb-0.05V-W-0.05Ti-0.3C-0.5Cu-0.02P	[45]
CAFA 7	Fe-25Ni-14Cr-2Mo-0.9Si-3.5Al-2Mn-1Nb-0.01V-W-0.1Ti-0.3C-0.5Cu-0.01P-0.01B	[46]
HTCAFA 4	Fe-35Ni-25Cr-Si-4Al-1Nb-0.3C-0.02P-0.01B+0.15Hf-0.03Y	[46,47]

Note: The names of AFA alloys are used in the literature.

**Table 2 materials-15-03515-t002:** The influencing elements to precipitates in AFA steels.

Precipitates	Influencing Elements
Carbides	Nb, Ti, V, Ta, Mn, P, W, C
B2-NiAl	Nb, Al, Cu, Si
Laves	Nb, W, Si and C
γ’ phase	Ni, Al, Zr, W, Cu
σ	Cr, Mo, Ti, Zr and V

**Table 3 materials-15-03515-t003:** Oxidation resistance of AFA steels in air.

Alloy	Condition			Weight Change (mg/cm^2^)
	Temperature (°C)	Water Vapor	Time (h)	
3-0.6 (Fe-20Ni-14Cr-3Al-0.6Nb) [20]	650	10%	10000	0.02
HC-2 (Fe-14Cr-12Ni-4.7Mn-2.5Al) [42]	650	10%	5000	0.2
HTUPS 4 (Fe-20Ni-14Cr-2.5Al-0.86Nb) [4]	800	10%	1000	0.09
13 (Fe-21Ni-14Cr-2.3Al-3Nb) [17]	800 (900)	dry 10%	500	0.05 (0.15) 0.045
AFA-SS-M (Fe-17Ni-15.3Cr-3.1Al-2.3Mo-9Mn) [18]	800	dry	1000	5
3-2.5 (Fe-20Ni-14Cr-3Al-2.5Nb) [20]	800	10%	5000	0.2
NF709-4 (Fe-25Ni-18Cr-3Al-0.8Nb) [65]	800	dry 10%	2000 800	0.088 0.075
OC4 (Fe-14Cr-25Ni-3.5Al-2.5Nb) [82]	800	10%	5000	0.3
CAFA4 (Fe-25Ni-14Cr-3.5Al-1Nb-2Mn-0.5Si-2Mo) [82]	800 700	10%	5000	0.25 0.1
22Cr-25Ni (Fe-25Ni-22Cr-2.5/3.5Al-0.45Nb) [33]	800	dry	120	0.13
AFA-NbTa (Fe-20Ni-18Cr-2.5Al-2.3Mo-0.45Nb-0.89Ta) [19]	800	dry	500	0.15
AFA + Al/C (Fe-25Ni-15Cr-4Al-2.5Nb-0.1C-0.01B) [16]	800	10%	8000	0.2
4 (Fe-35Ni-25Cr-4Al+Nb, C) [47]	1100	10%	1000	1
A (Fe-26Ni-14Cr-2.8Al-0.6Nb) [15]	900	dry	4000	2.5
AFA + B/C (Fe-25Ni-15Cr-3Al-2.5Nb-0.1C-0.107B) [16]	900	10%	1000	0.2

**Table 4 materials-15-03515-t004:** The short-term mechanical properties of AFA steels.

Name	YS (MPa)/UTS (MPa)/Elongation (pct)		Heat Treatment Condition
	Room Temperature	750 °C	
A-0.9	523/650/22	349/407/26	SA + 10% CW
DAFA 29(32ZCB)	560//22		As received
	1280//5.1		CR + 2.4 h Anneal
	1070//5.1		CR + 24 h
	800//5.1		CR + 240 h
	1150//6.2		SA + CR + 2.4 h
	1020//6.2		SA + CR + 24 h
	750//6.2		SA + CR + 240 h
AFA-Cu		700 °C	
(A1230)		394/450/20	Annealing (1230 °C) + 10% CW
B-1.0	261/613/51	201/357/32	SA
C-1.0	268/644/49	232/373/39	SA
AFA1	240/570/40	210/380/44	SA
	420/890/28	220/300/37	Aged (750 °C, 500 h)
AFA5	250/620/60	215/385/38	SA
	430/930/22	240/290/34	Aged (750 °C, 500 h)
AFA6	270/670/58	220/390/32	SA
	450/930/33	280/310/34	Aged (750 °C, 500 h)
NF709-4		330/490	SA
Fe-18Ni-12Cr based		700 °C	
	487/586/28	355/420/15	As received
	471/622/26.9	350/410/17.5	Aged (700 °C, 10 h)
	432/663/28.9	325/400/31.5	Aged (700 °C, 100 h)
	386/749/29.7	265/375/45.5	Aged (700 °C, 1000 h)
316 based AFA Dual phase steel	663/962/24.6		As received
	431/831/32		Aged (950 °C, 10 h)
	383/790/30		Aged (950 °C, 50 h)
	391/785/32		Aged (950 °C, 100 h)
AFA Fe-16Cr-3Al-2W-0.3Si-0.4Nb-0.04Y	592/779/27	700 °C 474/483/48	SA
Al-modified	205/338		SA,
	322/502		SA + Aged (800 °C, 2.4 h)
	362/707		SA + Aged (800 °C, 24 h)
	351/715		SA + Aged (800 °C, 240 h)
	383/736		SA + Aged (800 °C, 1325 h)
2.5Al AFA	573/703/27		As received
	595/781/22		Aged (700 °C, 500 h)
	581/741/19		Aged (700 °C, 3000 h)
Fe-14Cr-20Ni-2.5Al-1.5Nb-0.13Si-0.01Ti-2Mn-0.06C-2Mo-0.03W	638.9/774/41		HR
	526/698/53		HR (20%)
	715/843/35		HR (40%)
	757/849/25		HR (70%)
	834/924/7.3		HR (90%)
	494/757/37		Annealing
	455/680/49		Annealing +HR (20%)
	590/773/43		Annealing + HR (40%)
	684/828/29		Annealing + HR (70%)
	829/976/8.4		Annealing + HR (90%)
Fe-15Cr-25Ni-3Al-NbWCu	372/689/31.8		SA + Aged (700 °C, 72 h)
Fe-15Cr-25Ni-3Al-NbCu	454/824/31.8		Aged (700 °C, 72 h)
	593/841/41		As received
	731/916/28		Aged (700 °C, 2 h)
	1223/1310/26		Aged (700 °C, 20 h)
4Al-AFA	951/1140/23		Aged (700 °C, 100 h)
	1000/1184/25		Aged (700 °C, 500 h)
	978/1110/19		Aged (700 °C, 1000 h)
	636/1005/27		As received
	1153/1464/23		Aged (700 °C, 2 h)
4Al-2Cu-AFA	1137/1402/28		Aged (700 °C, 20 h)
	1075/1309/20		Aged (700 °C, 100 h)
	996/1225/17		Aged (700 °C, 500 h)
	914/1181/27		Aged (700 °C, 1000h)

SA: solution heat-treated at 1200 °C to 1250 °C. CR: Cold Rolling. HR: Hot Rolling.

**Table 5 materials-15-03515-t005:** The creep properties of AFA steels.

Composition	Creep Condition	Fracture Life (h)	Strengthening Phase
HTUPS 4	750 °C/170 MPa	2200 h	NbC
A-0.9	750 °C/170 MPa	357.6	NbC, Laves, B2
B-1.0	750 °C/170 MPa	337.1	NbC, Laves, B2, M_23_C_6_
C-1.0	750 °C/170 MPa	408.1	NbC, Laves, B2, M_23_C_6_
AFAW (Fe-20Ni-14Cr-2.5Al-Nb-0.16Si-0.1C-0.02V-2W)	700 °C/160 MPa	1598	Finer Laves
AFA-0	750 °C/150 MPa	1537	B2, Laves
AFA-Cu A1230	750 °C/150 MPa 700 °C/240 MPa 700 °C/150 MPa	2047 648 2821	Nano L12 containing Cu
NF709 base /0.5Nb0.08C0.1Ti	750 °C/100 MPa 750 °C/100 MPa	410 4500	Laves-Fe_2_Nb, sigma 10% cold work
NF709 base 1.5Nb0.15C	750 °C/100 MPa	230	Primary NbC, sigma
NF709 base 1.0Nb0.1C	750 °C/100 MPa	460	Primary NbC, Laves, sigma
CAFA 4	750 °C/100 MPa	3500	Fine L12
CAFA 6	750 °C/100 MPa	10326	Fine M_23_C_6_
WAFA	700 °C/160 MPa 700 °C/200 MPa	4100 1044 3119	Laves Pre-strain
HC-2	750 °C/100 MPa	484	(Ni, Fe)Al type B2, M_23_C_6_, α-Cu and NbC
32ZCB (DAFA 29)	750 °C/100 MPa	3008	L12, laves, B2
DAFA 26	760 °C/75 MPa	4921	Annealing at 800 °C 2.4 h

## Data Availability

Not applicable.

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
