# Peer review of "Research Progress of Alumina-Forming Austenitic Stainless Steels: A Review"

_materials, 2022, doi:10.3390/ma15103515_

Round 1

Reviewer 1 Report

Dear Authors,

I read carefully the manuscript entitled "Research Progress of Alumina-Forming Austenitic Stainless Steels: A Review" 

I find it well structured, containing useful and well synthesized information about alumina-forming austenitic stainless steels.

I have a few small remarks:

  1. Try to use bigger images so that the reader can easily understand what they are presenting;
  2. Can you reorganize Figure 4? As it is now, the images are very small. It is quite difficult to see them clearly;
  3. Figure 8: besides its dimension, you need to rewrite the text from the blue and red areas with a contrast color (i.e. white). The reader cannot read what is mentioned there;
  4. Table 4: Try to used the same notation (i.e. if you mention "0C" when mentioning temperatures, do it for all values in the table). I tried to found out what "CR" stands for. I didn't find out. Can you mention in the text or as a note to the table what "CR" means?
  5. Pay attention to the references. Check them and rewrite them according to the author's guide. (i.e. sometimes you mentioned the entire journal title and other times you mention it abbreviated etc.)

Kind regards 

Author Response

Dear editor, thank you for reviewing our manuscript in your busy schedule. First of all, thank you for your reasonable and constructive suggestions, which will greatly improve the quality of our paper. We have revised the paper according to your suggestions and requirements. Please check the attachment for the detailed modifications. Wish you success in your work!

Author Response

Dear editor, thank you for reviewing our manuscript in your busy schedule. First of all, thank you for your reasonable and constructive suggestions, which will greatly improve the quality of our paper. We have revised the paper according to your suggestions and requirements. Please check the attachment for the detailed modifications. Wish you!

Round 2

Reviewer 2 Report

Authors has been worked sincerely and improved the manuscript. I recommend to accept the article in Journal Materials.